# MHC Class I-Restricted TCR-Transgenic CD4^+^ T Cells Against STEAP1 Mediate Local Tumor Control of Ewing Sarcoma In Vivo

**DOI:** 10.3390/cells9071581

**Published:** 2020-06-29

**Authors:** Sebastian J. Schober, Melanie Thiede, Hendrik Gassmann, Carolin Prexler, Busheng Xue, David Schirmer, Dirk Wohlleber, Stefanie Stein, Thomas G. P. Grünewald, Dirk H. Busch, Guenther H. S. Richter, Stefan E. G. Burdach, Uwe Thiel

**Affiliations:** 1Department of Pediatrics, Children’s Cancer Research Center, Kinderklinik München Schwabing, School of Medicine, Technical University of Munich, 80804 Munich, Germany; melanie.thiede@tum.de (M.T.); hendrik.gassmann@tum.de (H.G.); carolin.prexler@tum.de (C.P.); busheng.xue@tum.de (B.X.); davidschirmer@t-online.de (D.S.); guenther.richter@charite.de (G.H.S.R.); stefan.burdach@tum.de (S.E.G.B.); 2Institute of Molecular Immunology/Experimental Oncology, Klinikum rechts der Isar, Technical University of Munich, 81674 Munich, Germany; dirk.wohlleber@tum.de; 3Max-Eder Research Group for Pediatric Sarcoma Biology, Institute of Pathology of the LMU, 80337 Munich, Germany; stefanie.stein@med.uni-muenchen.de (S.S.); thomas.gruenewald@med.uni-muenchen.de (T.G.P.G.); 4Division of Translational Pediatric Sarcoma Research, German Cancer Consortium (DKTK), German Cancer Research Center (DKFZ), 69120 Heidelberg, Germany; 5Institute of Pathology, Heidelberg University Hospital, 69120 Heidelberg, Germany; 6Institute for Medical Microbiology, Immunology and Hygiene, Klinikum rechts der Isar, School of Medicine, Technical University of Munich, 81674 Munich, Germany; dirk.busch@tum.de; 7Division of Oncology and Hematology, Department of Pediatrics, Charité—Universitätsmedizin Berlin, 13353 Berlin, Germany; 8German Cancer Consortium (DKTK), German Cancer Research Center (DKFZ), partner site Munich, 80336 Munich, Germany

**Keywords:** adoptive T cell transfer, CD4^+^ T cells, allorepertoire-derived TCR, Ewing sarcoma

## Abstract

In this study we report the functional comparison of T cell receptor (TCR)-engineered major histocompatibility complex (MHC) class I-restricted CD4^+^ versus CD8^+^ T cells targeting a peptide from *six transmembrane epithelial antigen of the prostate 1* (STEAP1) in the context of HLA-A*02:01. STEAP1 is a tumor-associated antigen, which is overexpressed in many cancers, including Ewing sarcoma (EwS). Based on previous observations, we postulated strong antitumor potential of tumor-redirected CD4^+^ T cells transduced with an HLA class I-restricted TCR against a STEAP1-derived peptide. We compared CD4^+^ T cell populations to their CD8^+^ counterparts in vitro using impedance-based xCELLigence and cytokine/granzyme release assays. We further compared antitumor activity of STEAP^130^-TCR transgenic (tg) CD4^+^ versus CD8^+^ T cells in tumor-bearing xenografted Rag2^−/−^γc^−/−^ mice. TCR tgCD4^+^ T cells showed increased cytotoxic features over time with similar functional avidity compared to tgCD8^+^ cells after 5–6 weeks of culture. In vivo, local tumor control was equal. Assessing metastatic organotropism of intraveniously (i.v.) injected tumors, only tgCD8^+^ cells were associated with reduced metastases. In this analysis, EwS-redirected tgCD4^+^ T cells contribute to local tumor control, but fail to control metastatic outgrowth in a model of xenografted EwS.

## 1. Introduction

Most pediatric solid malignancies, and some cancers of adults, are non-T cell-inflamed tumors. They are associated with low mutational burden, low numbers of neoantigens, an immunosuppressive microenvironment, and poor response rates to immune checkpoint blockade [1,2,3,4,5]. Pediatric sarcoma, in particular Ewing sarcoma (EwS), are aggressive malignancies of enigmatic histogenesis [6] with a high metastatic capacity. Survival rates drastically decrease when patients present with multifocal disease or early relapse [7,8]. Standard treatment strategies for EwS include surgery, irradiation, and high-dose chemotherapy followed by autologous stem cell transplantation (SCT). Hitherto, allogeneic SCT has not been proven to ameliorate overall survival of high-risk patients [9]. 

Adoptive transfer of transgenic (tg) cytotoxic CD8^+^ T cells targeting cancer/testis antigens, such as NY-ESO1, are currently evaluated in clinical trials with promising results [10,11]. *Six transmembrane epithelial antigen of the prostate 1* (STEAP1) is a tumor-associated antigen due to its high expression in several cancers, including EwS. It is physiologically expressed in prostate and urogenital tissues [12,13]. Moreover, STEAP1 is associated with an invasive and oxidative stress phenotype in EwS [14] and may be crucial for EwS homeostasis. This qualifies STEAP1 as a promising target of T cell-based immunotherapies [15]. 

Previously, we isolated a STEAP1^130^/HLA-A*02:01-peptide-restricted T cell receptor (TCR). Local tumor growth control of subcutaneously (s.c.) xenografted EwS tumor cells was significantly enhanced when treated with respective TCR tg CD8^+^ T cells in comparison to non-specific CD8^+^ T cells or peripheral blood mononuclear cells (PBMC) [16]. 

As in most healthy tissues and tumors, MHC class II expression is absent in EwS [17]. Immune cells, especially T cells and antigen-presenting cells (APC), are scarce in the tumor microenvironment of EwS. The most abundant immune cell population are tumor-associated macrophages (TAM) exhibiting features of immature/immunosuppressive (M0/M2) phenotypes [3,18,19,20]. 

Nowadays, it is widely accepted that the recruitment of CD4^+^ helper T cells is essential for sustained antitumor activity of cytotoxic CD8^+^ T cells or allograft rejection [21,22]. We previously observed strong tissue infiltration of CD4^+^ T cells in tumor-free Rag2^−/−^γc^−/−^ mice after adoptive transfer of EwS-redirected CD8^+^ T cells combined with PBMC, suggesting also a pivotal role of CD4^+^ T cells in tumor control in our model [23]. Thus, we sought to circumvent restrictions of absent HLA class II expression on tumors and the limited capacity of antigen cross-presentation of APC, by enabling CD4^+^ T cell to directly interact with tumor cells by introducing an MHC class I-restricted TCR. This would engineer a T cell combining CD4 and CD8 functions and provide both, help and kill. US Food and Drug Administration (FDA)- and European Medicines Agency (EMA)-approved anti-CD19-CAR-T cell products also contain both CD4^+^ and CD8^+^ T cells [24]. Thus far, only rare descriptions of MHC class I-restricted CD4^+^ T cells exist, with beneficial antitumor activities being observed when high-affinity TCRs were introduced in CD4^+^ T cells [25,26]. Furthermore, Xue et al. demonstrated increased antitumor immunity of CD4^+^ T cells when co-transduced with the CD8-receptor [27]. 

This study was conducted to examine the role of tumor-redirected MHC class I-restricted CD4^+^ in comparison to tumor-redirected CD8^+^ T cells against a STEAP1-derived HLA-A*02:01 restricted peptide.

## 2. Materials and Methods

### 2.1. Cell Lines

The cell lines used in this analysis were described previously [16]. EwS cell lines were cultured in RPMI 1640 medium (Life Technologies Limited, Paisley, UK) containing 10% fetal bovine serum (FBS) (Life Technologies Limited, Paisley, UK) and 100 U/mL penicillin, 100 μg/mL streptomycin (Life Technologies Corporation, Grand Island, NY, USA). The medium for LCL and T2 cells was additionally replenished with 1 mM Na-pyruvate and non-essential amino acids (both Life Technologies Limited, Paisley, UK). For culturing of IL-15-producing NSO and RD114 packaging cell lines, DMEM (Life Technologies Limited, Paisley, UK) containing 10% FBS, 1 mM Na-pyruvate, 1 mM non-essential amino acids and antibiotics mentioned above, were used. RD114 cells were a gift from Manuel Caruso (Center de Recherche en Cancérologie, Quebec, Canada) and NSO cells were a kind gift from S. Riddell (Seattle, Washington, USA). T cells were cultured in AIM-V medium (Life Technologies Limited, Paisley, UK) replenished with 5% human AB serum (SIGMA-ALDRICH CHEMIE GmbH, Steinheim, Germany) and antibiotics (see above), termed as T cell medium (TCM). All cell lines were routinely tested for purity and mycoplasma contamination as previously described [16].

### 2.2. Isolation of PBMC, CD4^+^ and CD8^+^ T Cell Populations

Healthy donor blood samples were purchased from DRK-Blutspendedienst (Baden-Wuerttemberg-Hessen, Ulm, Germany; obtained after IRB approval and informed consent). The study was conducted in accordance to the Declaration of Helsinki and approved by regulatory German regional government authorities (permission number: 50-8791-139.754.2122, date: 16.04.2015). PBMC were isolated by density-gradient centrifugation with Ficoll-Paque (GE Healthcare, Uppsala, Sweden)) according to supplier’s instructions. CD4^+^ and CD8^+^ T cells were positively isolated from PBMC using respective Dynabeads^®^ isolation kits according to manufacturer’s protocol (Thermo Fisher Scientific Baltics UAB, Vilnius, Lithuania).

### 2.3. Retroviral Transduction of T Cell Subsets, Purification, Expansion, and Culture Methods

The STEAP1^130^/HLA-A*02:01-specific TCR transgene was introduced into T cells using the retroviral vector pMP-71 [16]. Purified CD4^+^ and CD8^+^ T cells were activated with anti-CD3/CD28 Dynabeads™ (Thermo Fisher Scientific Baltics UAB, Vilnius, Lithuania) according to manufacturer’s recommendation and 100 U/mL recombinant human (rh) IL-2 (Novartis) and 2 ng/mL rh IL-15 (Bio-Techne, Minneapolis, MI, USA) for CD8^+^ T cells or 50 U/mL rh IL-2 and 5 ng/mL rh IL-7 (Bio-Techne, Minneapolis, MI, USA) for CD4^+^ T cells was added. Two days later, spin infection with virus supernatant from the packaging cell line RD114 was performed, as previously published [28]. On the next day, T cells were re-infected with the same approach. After verifying successful transduction via FACS staining, T cell subsets were purified with anti-phycoerythrin (PE) magnetic beads (Miltenyi Biotec, Bergisch Gladbach, Germany) coupled to STEAP1^130^/HLA-A*02:01-specific PE labeled multimers (in-house production), as described previously [29]. Detailed assessment of STEAP1^130^/HLA-A*02:01-specific TCR functionality and potential cross reactivity was published previously [30].

### 2.4. FACS Staining and Analysis

The phenotype and status of TCR-tg T cells were evaluated via surface on a FACS Calibur (BD Biosciences, Heidelberg, Germany), as previously described [16,28,31]. Shortly, cell staining for TCR-specificity was performed with STEAP1^130^/HLA-A*02:01-specific and irrelevant peptide/HLA-A*02:01-PE labeled multimers (STEAP1_M and IRR_M), CD4- or CD8-FITC (fluorescein isothiocyanate). In all stainings isotype FITC-, PE- and APC- (allophycocyanin) labeled IgG mAb served as negative controls. Phenotyping was performed with epitope-specific FITC-, PE-, APC-labeled mAb as indicated in the text (purchased from BD Bioscience, Heidelberg, Germany or Miltenyi Biotec, Bergisch Gladbach, Germany). Measured events were analyzed using the FlowJo™ 10.1 software (BD Bioscience, Heidelberg, Germany).

### 2.5. Functional Characterization of STEAP1^130^/HLA-A*02:01-Specific TCR Transgenic T Cell Subsets

T cell-mediated cytotoxicity was monitored with the impedance-based xCELLigence assay (Roche Diagnostics, Penzberg, Germany), allowing continuous measurement of T cell activity against target cell lines, such as A673 and SK-N-MC (in different effector-to-target ratios). Untreated tumor cells were used as controls. Moreover, 10 × 10^3^ A673 and 20 × 10^3^ SK-N-MC tumor cells were plated at the beginning of experiment and therapeutic cells were added at cell indexes of 1–2 (after a period of cell adherence of at least 18 h). T cell specificity was confirmed with interferon-γ (IFNγ) and granzyme B ELISpot assays (Mabtech AB, Nacka Strand, Sweden), according to manufacturer’s information. Therefore, effector and target cells were co-cultured for 20 h at 37 °C, 5% CO_2_, at a ratio of 1:20 (if not indicated otherwise) and analyzed with EliSpot Reader (machine and software version 5.0; Advanced Imaging Devices GmbH, Straßberg, Germany. 

For detailed analyses of chemokine and cytokine release patterns from supernatant of co-cultured T cells with tumor cells, a 48-plex panel was performed according to supplier’s instructions (Bio-Rad, Munich, Germany). Supernatants were collected from xCELLigence experiments at indicated time points (triplicates from one condition were pooled), centrifuged, and stored at −80 °C for further analyses. For heatmap generation, RStudio software version 1.0.143 (RStudio, Inc., Boston, MA, USA) and the *heatmap.2* script was used [32].

### 2.6. Animal Model

Immuno-compromised Rag2^−/−^γc^−/−^ mice (BALB/c background), obtained from the Central Institute for Experimental Animals (Kawasaki, Japan) were maintained in our animal facility under pathogen-free conditions after approval of local regulatory authorities according to German Federal Law and in accordance with institutional guidelines (permission number: 55.2-2532.Vet_02-15-103, date: 21.10.2015). Experiments were performed in 10–30-week old mice.

### 2.7. In Vivo Experiments

#### 2.7.1. Local Tumor Control

To examine in vivo local tumor control, the previously published animal setting was reused [16]. Briefly, 2 × 10^6^ A673 EwS cells (partly luciferase expressing, when indicated) were injected subcutaneously (s.c.) in the lower back of mice. Once the tumor was palpable, mice were randomized into indicated treatment groups. At day 3, mice were irradiated (3.5 Gy) and received (1) 1 × 10^7^ non-specific PBMC, (2) 5 × 10^6^ tgCD4^+^ T cells together with 5 × 10^6^ CD4-depleted non-specific PBMC, (3) 5 ×1 0^6^ tgCD8^+^ T cells together with CD8-depleted non-specific PBMC, or (4) 2.5 × 10^6^ tgCD4^+^, 2.5 × 10^6^ tgCD8^+^ T cells together with 5 × 10^6^ non-specific PBMC intraperitoneally (i.p.) on the next day. All mice received i.p. 1 × 10^7^ irradiated (80 Gy) IL-15-producing NSO cells biweekly until the end of experiment at day 17. Then, mice were sacrificed; s.c. tumors were explanted and weighed. Blood, spleen, and bone marrow were subjected to red-blood-cell lysis buffer (Invitrogen) according to manufacturer’s recommendations, stained with human anti-CD4/CD8, irrelevant/specific multimers, and negative controls, as mentioned above, before flow cytometric assessment.

#### 2.7.2. Experimental Metastasis

In order to study T cell-mediated control of metastatic organotropism, we applied the previously published model of experimental metastasis [28]. Herein, mice were irradiated on day −1 (3.5 Gy), before administering 2.5 × 10^6^ A673 EwS cells via tail vein injection. Then, mice received the same cell numbers and treatment conditions (1–4) as mentioned above, as well as biweekly injections of irradiated IL-15-producing NSO cells. Moreover, 30 days after tumor/therapeutic cell injection, mice were sacrificed and organs (spleen, lung, and liver) were formalin-fixed for histopathological analyses.

#### 2.7.3. Bioluminescence Monitoring of Implanted Tumors

Total photon flux (maximum; 3–10 min after luciferin administration) in local tumor control model was measured using an IVIS Lumina LT-Series III instrument (PerkinElmer LAS, Rodgau, Germany) at indicated timepoints after administration of 150 mg luciferin/kg bodyweight i.p. (PerkinElmer LAS, Rodgau, Germany).

### 2.8. Histopathological Analysis

Liver and lungs from mice (experimental metastasis model) were paraffin-embedded and stained for hematoxylin/eosin (HE). Percentage of tumor spread in respective organ was assessed and calculated from representative slides in organ overview (4× magnification).

### 2.9. Statistical Analysis

Normally distributed, quantitative values were analyzed via two-tailed student’s t-test regarding mean, standard deviation (SD), and SD of the mean (SEM). Arbitrarily distributed values were analyzed with Kruskal–Wallis and Mann–Whitney U test; linear regression modeling was computed for tumor weight, human CD4^+^, and CD8^+^ T cell frequencies in blood of respective animals; all done with Prism 5 (GraphPad Software, San Diego, CA, USA); *p* values < 0.05 were considered statistically significant (* *p* < 0.05; ** *p* < 0.005; *** *p* < 0.0005).

### 2.10. Analysis of Public Patient Data 

We analyzed publicly available data of EwS patients for expression of subtype specific T cell marker genes (CD4 and CD8alpha) and expression of the *C-X-C Motif Chemokine Ligand 9/10* (CXCL9/10)-*C-X-C Motif Chemokine Receptor 3* (CXCR3)-axis transcripts. For GSE63155 (*n* = 46), we conducted a Kaplan–Meier analysis with regard to overall and event-free survival. The raw expression data and survival data was downloaded from the GEO database and analyzed using R [32]. Expression data was normalized; background corrected using RMA and annotated using *brainarray CDF* (ENTREZG). The survival analysis was conducted using the packages *survival* and *survminer*. All genes of interest were tested separately. A median cutoff for expression levels was used to assign patient groups. Two other datasets (Savola dataset, GSE17618, *n* = 44; Francesconi dataset, GSE12102, *n* = 37) were used to compare the expression of our genes of interest with regard to status (primary vs. metastasis vs. recurrence). For each gene, we compared the expression in the three groups by applying ANOVA and generating boxplots using R2 [33]. Note that the Savola dataset was filtered for sample series id (GSE17618) and diagnosis (Askin tumor, EwS, PNET).

## 3. Results

### 3.1. In Vitro Antitumor Activity of tgCD4^+^ T Cells Increases Over Time 

Peripheral blood HLA-A*02:01-negative CD4^+^ T cells and CD8^+^ T cells from healthy donors were successfully transduced with a STEAP1^130^/HLA-A*02:01-specific TCR and purified via STEAP1^130^/HLA-A*02:01-specific PE-labeled multimer positive selection (Figure 1A,B). 

T cell-mediated cytotoxicity, measured by contact-dependent detachment of adherent tumor cells [34], of STEAP1-specific CD4^+^ T cells (tgCD4) was initially observed after 46 days of culture (Figure 1C). Furthermore, a donor-dependency concerning antitumor activity of tgCD4^+^ T cells was observed (Figure 2A), showing complete detachment (i.e., killing) of tumor cells at effector-to-target ratios (E:T) 20:1 to 5:1. Compared to STEAP1-specific CD8^+^ T cells (tgCD8), antitumor effects of tgCD4 T cells increased with the number of days in cell culture (Figure 2B). Despite increased antitumor activity of tgCD4 T cells, tgCD8 T cells always showed superior antitumor activity at evaluated effector-to-target ratios at all points in time. Moreover, no additive or synergistic effects of tgCD4 T cells, when applied together with tgCD8 T cells (at a ratio of 1:1) was observed (Figure 2C,D).

To further characterize the tgCD4 T cell-mediated antitumor responses, we performed IFNγ and granzyme B release assays. Here, we observed similar IFNγ release patterns of tgCD4 and tgCD8 T cells when co-cultured with STEAP1^130^-peptide-loaded T2 cells, indicating high STEAP1^130^-peptide/HLA-A*02:01 specificity at day 46 (Figure 3A). Furthermore, specific IFNγ (Figure 3B) and dose-dependent granzyme B release (Figure 3C) of tgCD4 T cells was triggered when co-cultured with HLA-A2 positive target cells; although lower than for tgCD8 T cells, as previously published [16] (all assessed in donor 1). T cell phenotyping of tgCD4 and tgCD8 T cells was performed weekly to exclude significant differences of exhaustion or de-differentiation of tgCD4 T cells towards regulatory T cell or T helper 2 phenotypes. Actually, more tgCD4 than tgCD8 T cells exhibited a CD45RO^+^/CD62L^+^ central memory phenotype, indicating prolonged in vitro fitness. Exhaustion markers were comparable and extracellular analysis of chemokine receptors were indicative for a T helper 1 phenotype (Appendix A) [35]. 

When examining the supernatants at the end of xCELLigence co-culture experiments with a cytokine multiplex assay, antitumor activity of tgCD4 T cells (E:T = 20:1) was associated with an increase of CXCL9, CXCL10, but without a significant increase of other effector cytokines, such as IFNγ, IL-4, IL-10 or IL-17A (Appendix A). Whereas IFNγ and tumor necrosis factor alpha (TNF) were the most elevated effector cytokines in tgCD8 T cell-containing settings (also tgCD4 plus tgCD8 T cells), as expected. In general, there was a tendency for higher CXCL9/10 levels within tgCD4^+^ T cell-containing settings in all analyzed supernatants (Appendix A), hinting to both possible antitumor and tumor escape mechanisms mediated by the CXCL9/10-CXCR3 axis.

### 3.2. STEAP1^130^/HLA-A*02:01-Specific TCR Transgenic CD4^+^ T Cells Control Tumor Growth in a Xenografted Local Tumor Model

The tumor weights at the end of experiments (mean and median tumor weight) of all animals receiving STEAP1^130^/HLA-A*02:01-specific TCR transgenic T cells (tgCD4, tgCD8, and tgCD4 plus tgCD8 T cells) were lower compared to the non-specific PBMC control group. No significant variation of medians was observed in-between groups when analyzed with Kruskal–Wallis test (*p* = 0.0976). Mann–Whitney U test comparison of control group versus each study group respectively, revealed a statistically significant lower median tumor weight for tgCD4 and tgCD4/tgCD8 groups (Figure 4A). 

#### 3.2.1. Bioluminescence-Based Analysis of Tumor Burden in Local Tumor Model

Tumor activity assessment via bioluminescence in general supported the findings obtain from analysis of tumor weights. Although, the mean photon flux_max_ at the end of experiments, indicated a significant superiority in controlling local tumors for animals receiving tgCD8^+^ and tgCD4^+^ plus tgCD8^+^ T cells (Figure 4B). Interestingly, when comparing tumor control at day 6 after therapeutic cell application, tgCD4 study group animals showed a significant lower fold change in mean photon flux_max_ (day 10 versus day 4) compared to tgCD8 study group animals, possibly indicating different dynamics in antitumor activity of tgCD4 and tgCD8 T cells (Appendix A).

#### 3.2.2. Correlation of T Cell Frequencies in Blood and Spleen with Tumor Weight

At the end of the experiment, human T cell frequencies in bone marrow (not shown) and spleen were lower than 1% within the gated lymphocyte population. Human T cell frequencies in the peripheral blood were lower than 10% within the gated lymphocyte population with highest frequencies observed in the tgCD4/tgCD8 study group animals. TgCD4 and tgCD8 T cells, identified via peptide/MHC-specific pentamer staining could not be detected definitely in all examined organs/tissues (i.e., tumor, bone marrow, spleen, blood; not shown).

Interestingly, the percentage of human CD4^+^ T cells (with unknown specificity) in blood within the respective gate, were significantly higher in analyzed animals receiving tgCD8^+^ T cells (i.e., tgCD8 and tgCD4/tgCD8 group; Figure 4D left). Human CD8^+^ T cell frequencies in blood did not differ in all analyzed animals significantly; with the highest percentage of CD8^+^ T cells observed in animals within the tgCD4/tgCD8 group (Figure 4D right). Although, very low T cell frequencies were detected in spleens, the percentage of CD8^+^ T cells were significantly higher within the tgCD4/tgCD8 compared to all other groups (Figure 4E).

Next, we sought to examine a possible correlation of tumor weight at the end of the experiment and T cell frequencies. Whereas correlation analysis did not suggest an association of tumor weight and T cell frequencies in spleen, higher CD4^+^ and CD8^+^ T cell frequencies in the blood of animals showed a significant and negative correlation with tumor weight (not shown). Linear regression modeling only revealed a significant deviation of the slope from zero for blood CD4^+^ T cells frequencies (R^2^ = 0.2958, *p* = 0.024; Figure 4C).

### 3.3. STEAP1^130^/HLA-A*02:01-Specific TCR Transgenic CD4^+^ T Cells Do Not Control Metastatic Outgrowth in a Model of Experimental Metastasis

A beneficial effect of local tumor control was observed in all animals receiving STEAP1^130^/HLA-A*02:01-specific TCR transgenic T cells (tgCD4, tgCD8, and tgCD4 plus tgCD8 T cells) in comparison to mice receiving non-specific PBMC. In a second step, we assessed control of metastatic organotropism in our model of experimental metastasis by tail vein injection of tumor cells and therapeutic cells. Here, we observed significantly more liver metastases in treatment groups of animals receiving STEAP1-specific TCR-transgenic CD4^+^ T cells (tgCD4 and tgCD4/tgCD8 study groups; Figure 5A and Appendix A). Control of pulmonary metastasis did not differ significantly in all study groups animals, although a similar tendency of metastatic outgrowth was observed, compared to liver metastases (Figure 5B).

### 3.4. Exploratory Bioinformatic Analysis Reveals a Correlation of CXCL10 and Survival and Points to Differences of T Cell Subset-Specific Transcripts Depending on Primary, Metastatic or Relapsed Site

To screen for further evidence concerning contributing roles of CD4^+^ T cells and the CXCL9/10-CXCR3-axis in EwS disease control or pathogenesis, we analyzed publicly available transcriptome datasets. In GSE63155, high CXCL10 expression significantly correlated with poor event-free and overall survival (Appendix A). When comparing CD4 and CD8A transcripts, ANOVA analysis of 2 EwS datasets confirmed a tendency that CD4 transcripts are higher expressed in primary tumors than in relapsed or metastatic sites, whereas CD8A transcripts are higher expressed in metastatic compared to relapsed or primary sites (Appendix A). These finding might link our results from both in vivo studies and in vitro analyses of cytokines, hinting to a possible dominance of CD4^+^ T cells to mediate local disease control in EwS (e.g., by induction of CXCL9/10 in the tumor microenvironment).

## 4. Discussion

In this work, we sought to elaborate T cell-mediated antitumor effects of tumor-redirected MHC class I-restricted CD4^+^ T cells in comparison to tumor-redirected CD8^+^ T cells and inquired possible synergistic effects of both subsets in our in vitro and in vivo models.

We demonstrated the potential of genetically engineered CD4^+^ T cells to express an MHC class I-restricted (CD8^+^ T cell-derived) TCR against the EwS tumor antigen STEAP1^130^. Functional in vitro comparison to CD8^+^ counterparts revealed increasing antitumor activity with the time of culturing with a cytokine cocktail containing IL-2 and IL-7. At 46 days of culturing, tgCD4 T cells released comparable levels of effector cytokines, when co-cultured with *STEAP1^130^/HLA-A*02:01*-expressing target cells in ELISpot analyses. Nonetheless, in vitro antitumor activity indicated inferior antitumor potential of tgCD4 compared to tgCD8 T cells. Neither direct additive nor synergistic effects of tgCD4 and tgCD8 T cells were observed in this model. This is in contrast to observations from Matsuzaki et al. (2015), who confirmed a direct helping effect (antigen-presenting cell-independent) of NY-ESO-1-specific CD4^+^ T cells to strengthen antitumor efficacy of CD8^+^ T cells in vitro and in vivo, which was mediated by MHC class II-specific recognition and the consecutive induction of a tumor growth arrest [36]. Furthermore, donor-dependent variations observed in in vitro co-culture experiments (Figure 2A) appeared to be a general limitation to produce homogenous CD4^+^ therapeutic T cells with the herein presented approach. Moreover, the proliferative potential and in vivo persistence of adoptively transferred tgCD4 and tgCD8 T cells was not addressed in this study.

As Xue et al. (2013) reported increased antitumor activity of tumor-redirected CD4 T cells when co-transduced with the CD8-receptor, we cannot exclude that observed limitations of our tgCD4 T cells could be attributed to the lack CD8-co-expression [27]. STEAP1^130^/HLA-A*02:01-TCR affinity might also be taken under consideration, as introducing high-affinity or affinity-matured TCRs were shown to augment CD4^+^ T cell-mediated antitumor activity [25,26]. 

Nevertheless, we observed an antitumor effect of tgCD4 T cells in our in vivo model of local tumor growth control, which is in line with several other reports focusing on the contribution of CD4^+^ T cells to control s.c.-implanted tumors. Contributing mechanisms included CD4^+^ T cell-mediated induction of CXCL9 and CXCL10 in the tumor microenvironment to attract CXCR3-expressing CD8^+^ T cells or to confer a specific cytotoxic CD8^+^ T cell effector program amongst others [21,37]. Bioluminescence monitoring suggested a superior tumor control of tgCD4 study group animals at an earlier point in time (e.g. 6 days after T cell application), concurring with results of aforementioned reports. Furthermore, the discrepancy of tumor weight and bioluminescence observed in the tgCD4 study group might suggest that xenografted tumors were not killed, but underwent changes associated with senescence or dormancy. Immune cell-mediated induction of tumor dormancy or senescence is well known [38]. Müller-Hermelink et al. (2008) described a dual role of CD4^+^ T cells, which either control or enhance multistage carcinogenesis depending on the presence or absence of either IFNγ or tumor necrosis factor receptor 1 (TNFR1) signaling; partly mediated by CXCL9/10 induction in the tumor microenvironment [39]. 

This is in line with Berghuis et al. (2009) who associated CXCL9/10 expression in EwS tumor and stromal cells with higher numbers of CXCR3^+^CD3^+^ T cells [18]. A recent study comparing transcriptome data of disease stabilization and tumor progression from a neuroblastoma patient after oncolytic virotherapy found that CXCL9/10-CXCR3 levels were significantly upregulated during tumor progression, underlining the dichotomous role of this axis, which was also shown to promote metastasis when active as an autocrine feedback-loop [40,41].

Hence, we could speculate that our in vivo results (of CD4^+^ T cell-mediated local tumor control but failure to control metastases) reflect aforementioned phenomena due to the absence of strong and prolonged IFNγ and TNF signaling despite the induction of CXCL9/10 (Appendix A).

Results from this study point out the possibility to genetically engineer and redirect both CD4^+^ and CD8^+^ T cells resulting in EwS recognition and specific killing. However, they suggest careful evaluation of therapeutic CD4^+^ T cell products, in regard to donor-dependent variations, as well as TCR affinity and functional avidity testing, in order to avoid possible tumor growth promoting effects.

## Figures and Tables

**Figure 1 cells-09-01581-f001:**
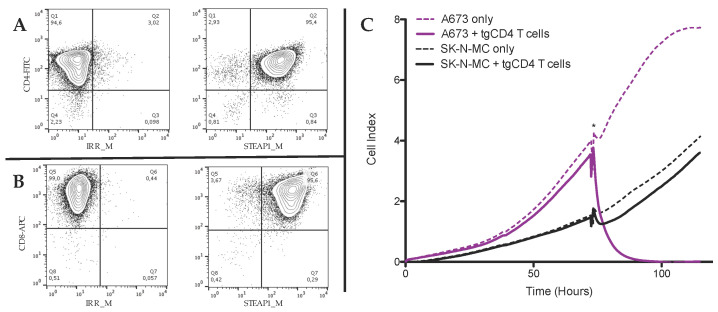
STEAP1^130^/HLA-A*02:01-specific TCR transgenic CD4^+^ (tgCD4) T cells and CD8^+^ (tgCD8) T cells of high purity were generated and tgCD4 cells showed specific antitumor activity in vitro. Contour plots of (**A**) STEAP1-specific tgCD4 cells, co-stained with CD4-FITC, irrelevant multimer-PE (IRR_M), relevant multimer-PE (STEAP1_M), and (**B**) STEAP1-specific tgCD8 cells, co-stained with CD8-APC, IRR_M, and STEAP1_M. (**C**) Contact dependent growth of A673 (HLA-A*02:01-positive) or SK-N-MC (HLA-A*02:01-negative) tumor cells was monitored in xCELLigence assay at effector-to-target ratio 20:1 at day 46 of tgCD4 T cell culture. Addition of T cells/mock to pre-cultured tumor cells is asterisked (*). Specific target recognition by T cells resulted in growth inhibition (detachment) of A673 (purple) but not of SK-N-MC (negative control) in comparison to mock-treated A673 or SK-N-MC cells (dashed lines, respectively).

**Figure 2 cells-09-01581-f002:**
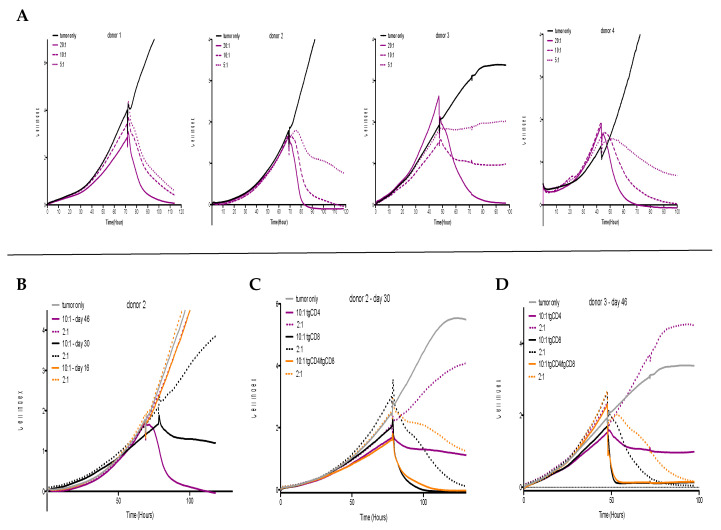
Donor- and time-dependent antitumor activity of STEAP1^130^/HLA-A*02:01-specific TCR transgenic CD4^+^ (tgCD4) T cells and CD8^+^ (tgCD8) T cells in vitro. (**A**) STEAP1^130^ tgCD4 T cells show donor-dependent antitumor activity after 6 weeks of expansion and culture against A673 cells (4 donors). (**B**) Antitumor activity of tgCD4 T cells increased with the number of days in cell culture; assessed at day 16, 30, and 46 in effector-to-target ratios (E:T) 10:1 and 2:1. (**C**,**D**) Evaluation of antitumor activity of tgCD4, tgCD8, and tgCD4/tgCD8 (1:1) T cells at day 30 and day 46, indicating the superiority of tgCD8 T cells at all assessed points and E:T ratios (i.e., 10:1 and 2:1), respectively.

**Figure 3 cells-09-01581-f003:**
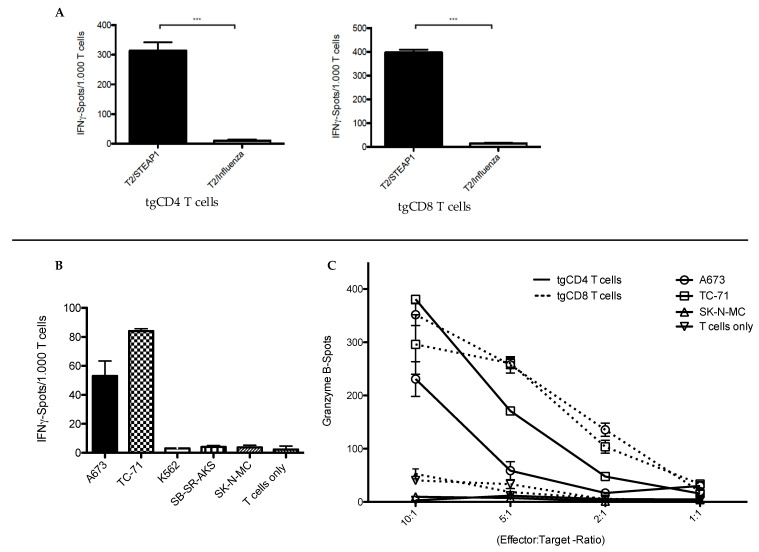
Peptide specificity and effector cytokine release of STEAP1^130^/HLA-A*02:01-specific TCR transgenic CD4^+^ (tgCD4) and CD8^+^ (tgCD8) T cells in vitro. (**A**) IFNγ ELISpot analysis, comparing tgCD4 and tgCD8 T cells (1.000 T cells per 20.000 peptide-loaded T2 cells (STEAP1^130^- and influenza control-peptides). (**B**) IFNγ-ELISpot analysis of tgCD4 T cells when co-cultured with HLA-A*02:01-positive (A673, TC-71) and HLA-A*02:01-negative target cells (1.000 T cells per 20.000 target cells). (**C**) Granzyme B-ELISpot analysis of tgCD4 and tgCD8 T cells at different effector-to-target ratios (20.000 target cells). Error bars represent the standard deviation of triplicate experiments. Asterisks indicate significance levels. * *p* < 0.05; ** *p* < 0.005; *** *p* < 0.0005; ns = not significant.

**Figure 4 cells-09-01581-f004:**
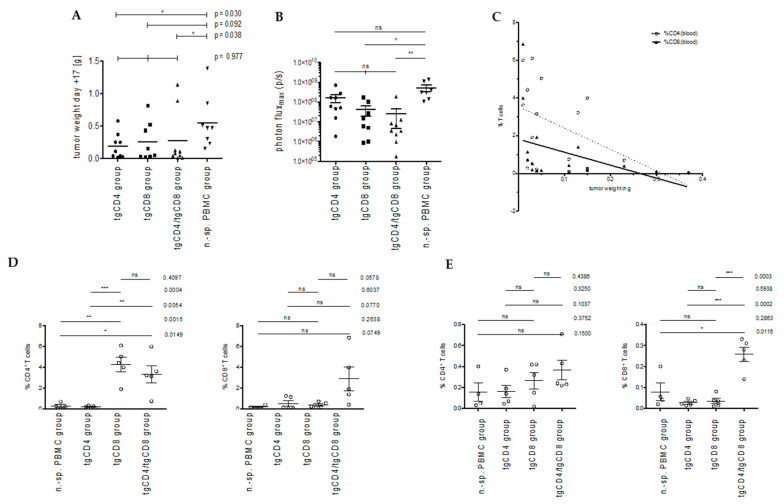
Local tumor control and human T cell frequencies in blood in vivo. Antitumor activity in treatment groups (tgCD4, tgCD8, tgCD4/tgCD8, and n.-sp-PBMC) the end of experiment (day +17) indicated as tumor weight (**A**) and bioluminescence (**B**), each dot corresponds to an animal. (**C**) Linear regression modeling of tumor weight, CD4^+^ and CD8^+^ T cell frequencies in blood of corresponding animals. Human T cell frequencies (%) in specific lymphocyte gate, depicted for CD4^+^ (left) and CD8^+^ T cells (right) in blood (**D**) and spleen (**E**) in respective treatment groups (*n* = 5 per group). Error bars represent the SEM. Asterisks indicate significance levels. * *p* < 0.05; ** *p* < 0.005; *** *p* < 0.0005; ns = not significant.

**Figure 5 cells-09-01581-f005:**
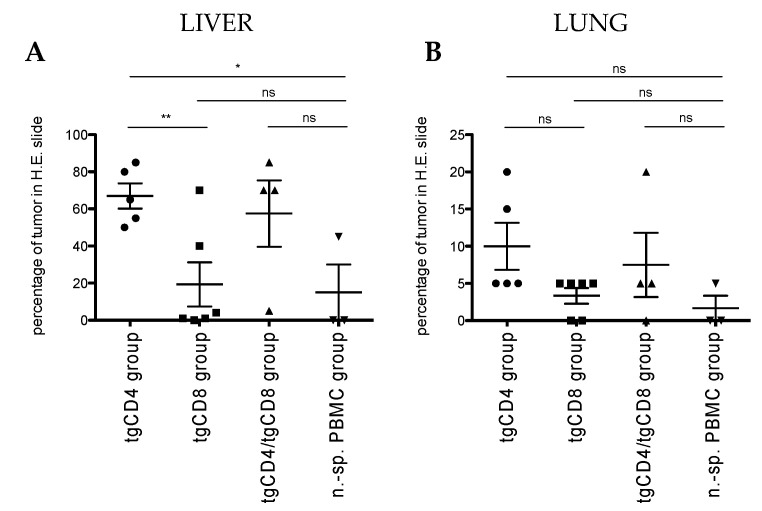
Control of metastatic outgrowth in a model of experimental metastasis. In vivo antitumor activity of STEAP1^130^/HLA-A*02:01-specific TCR transgenic T cells (tgCD4, tgCD8, and tgCD4 plus tgCD8 T cells - 1:1) compared to non-specific (n.-sp.) PBMC measured by control of metastatic outgrowth in liver **(A)** and lung **(B)** 30 days after tumor/therapeutic cell injection in a model of metastatic organotropism. Each dot corresponds to one animal and the percentage of tumor spread within respective organ. Error bars represent the SEM. Asterisks indicate significance levels. * *p* < 0.05; ** *p* < 0.005; *** *p* < 0.0005; ns = not significant.

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
