# Peer review of "MHC Class I-Restricted TCR-Transgenic CD4+ T Cells Against STEAP1 Mediate Local Tumor Control of Ewing Sarcoma In Vivo"

_cells, 2020, doi:10.3390/cells9071581_

Round 1
Reviewer 1 Report
The manuscript by Schober et al reports results on the functional in vitro and in vivo comparison of MHC class I restricted CD4+ T-cells versus CD8+ T-cells after gene transfer of a TCR specific for STEAP-1, a cancer/testis antigen. The manuscript is interesting and well written. Experimental design is sound. The authors are experienced in the field.
Although the idea of transferring class I specific TCR into CD4+ cells is not new, the authors bring relevant contribution to the field by clearly showing that, in their model, the activity of redirected CD4+ T-cells is similar to CD8+ counterparts in vitro and in vivo for local tumor control, but inferior in controlling metastatic outgrowth.
I have some minor comments:
- Which is the proliferative potential of cultured Tg cells in vivo? Although cells cultured for 46 days did not display markers of exhaustion and/or signs of terminal differentiation, they might, for instance, have a reduced telomer length impairing their in vivo proliferation
- Is variability amongst donors true also for CD8+ T-cells?
- Which is the functional avidity of redirected CD4+ T cells compared to CD8+ counterparts? Peptide titration experiments on T2 cells might be helpful
- Figure 3, panel B and C: activity of CD8+ cells is lacking for comparison
- Were there enough recovered human cells from mice to examine their phenotype? Did recovered T cells show any sign of exhaustion or differentiation switch?
- Legends to supplemental figures can be improved
- Reference for the xCELLigence assay is missing

Reviewer 2 Report
In this work the authors test the antitumor capacity of CD4+T cells expressing an MHC class I restricted TCR against EWS tumor antigen STEAP1. A limitation of this model as mentioned by the authors is the lack of CD8 expression by these T cells.
The authors show in in vitro studies that the tgCD4+ T cells have a antitumor activity however tgCD8 T cells showed superior antitumor activity. In addition it was tested the in vivo activity of these cells in xenografted local tumor model. The in vivo results were contradictory as the tgCD4+ T cells induce decreased tumor weight but did not have an effect on tumor burden when the analyzes was done by bioluminescence (figure 4). In addition tgCD4+ T cells do not control metastatic outgrowth.
Detailed comments:
Line 110 FACs should be written FACS
Line 180 CD8A should be CD8alpha
The statistic description of the methods is incomplete as the test Kruskal-Wallis is used in the results section and not mention in the methods.
Line 205 where is written Figure 2D should be written Figure 2C,D
The results described in Figure 1B;C;D correspond to one donor the experiments should be repeated with additional donors. As no conclusions could be draw from one experiment. Specially as the authors state that there is a donor dependency concerning antitumor activity of tgCD4+ T cells.
In Fig.3 legend should be state how many experimental replicates of each experiment was done.
For Supplemental Figure S1 a representative example of the flow cytometry gating strategy should be added.
Line 215 and 216 state that “analysis of transcription factors confirmed a T helper 1 phenotype (Supplemental Figure S1 and S2).” However Figure S2 only represent one donor so no conclusion can be made. The authors should remove the sentence or perform the same analyzes in more donors.
In Figure 4 and 5 the authors should only mention the significance (p value) for the significant differences if you not show it is assumed that there is no difference.
In the result sections combine the results from line 251 to 287 in the same section as they refer two too methods of accessing the effect of transgenic T cells in tumor control. (All the results are shown in Figure 4).
From line 269 to 270:
The authors refer that “At the end of the experiment, human T cell frequencies in bone marrow (not shown) and spleen were lower than 1 % within the gated lymphocyte population”. The authors are using Rag2-/-gc-/- (so no T cells, B cells or NK cells) and mice were injected with T cells + PBMC. It is puzzling that lower than 1% of the cells in the lymphocyte gate are T cells. It will be important to show a plot of flow cytometry gating strategy and ideally the characterization of the other cells in the lymphocyte gate.
Supplemental table 1 should be removed.
Supplemental figure 5 should be integrated within fig 4, as supplemental figure 5 refers to spleen and figure 4 blood. The correlation should also be shown in the main manuscript as it is the title of the section.
Round 2
Reviewer 2 Report
I agree with the alterations made